# GAUGE EQUIVARIANT MESH CNNS
## ANISOTROPIC CONVOLUTIONS ON GEOMETRIC GRAPHS

**Pim de Haan**[*]
Qualcomm AI Research[†]
University of Amsterdam

**Maurice Weiler**[*]
QUVA Lab
University of Amsterdam

**Taco Cohen**
Qualcomm AI Research

**Max Welling**
Qualcomm AI Research
University of Amsterdam

## ABSTRACT

A common approach to define convolutions on meshes is to interpret them as a graph and apply graph convolutional networks (GCNs). Such GCNs utilize *isotropic* kernels and are therefore insensitive to the relative orientation of vertices and thus to the geometry of the mesh as a whole. We propose Gauge Equivariant Mesh CNNs which generalize GCNs to apply *anisotropic* gauge equivariant kernels. Since the resulting features carry orientation information, we introduce a geometric message passing scheme defined by parallel transporting features over mesh edges. Our experiments validate the significantly improved expressivity of the proposed model over conventional GCNs and other methods.

## 1 INTRODUCTION

Convolutional neural networks (CNNs) have been established as the default method for many machine learning tasks like speech recognition or planar and volumetric image classification and segmentation. Most CNNs are restricted to flat or spherical geometries, where convolutions are easily defined and optimized implementations are available. The empirical success of CNNs on such spaces has generated interest to generalize convolutions to more general spaces like graphs or Riemannian manifolds, creating a field now known as geometric deep learning (Bronstein et al., 2017).

A case of specific interest is convolution on *meshes*, the discrete analog of 2-dimensional embedded Riemannian manifolds. Mesh CNNs can be applied to tasks such as detecting shapes, registering different poses of the same shape and shape segmentation. If we forget the positions of vertices, and which vertices form faces, a mesh $M$ can be represented by a graph $\mathcal{G}$. This allows for the application of *graph convolutional networks* (GCNs) to processing signals on meshes.

---

[*]Equal Contribution
[†]Qualcomm AI Research is an initiative of Qualcomm Technologies, Inc.

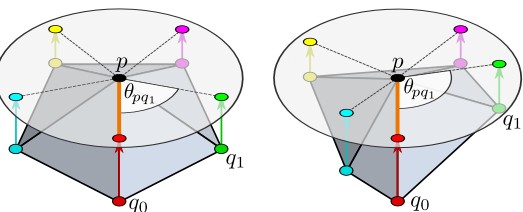

Figure 1: Two local neighbourhoods around vertices $p$ and their representations in the tangent planes $T_pM$. The distinct geometry of the neighbourhoods is reflected in the different angles $\theta_{pq_i}$ of incident edges from neighbours $q_i$. Graph convolutional networks apply isotropic kernels and can therefore not distinguish both neighbourhoods. Gauge Equivariant Mesh CNNs apply anisotropic kernels and are therefore sensitive to orientations. The arbitrariness of reference orientations, determined by a choice of neighbour $q_0$, is accounted for by the gauge equivariance of the model.

However, when representing a mesh by a graph, we lose important geometrical information. In particular, in a graph there is no notion of angle between or ordering of two of a node's incident edges (see figure 1). Hence, a GCNs output at a node $p$ is designed to be independent of relative angles and *invariant* to any permutation of its neighbours $q_i \in \mathcal{N}(p)$. A graph convolution on a mesh graph therefore corresponds to applying an *isotropic* convolution kernel. Isotropic filters are insensitive to the orientation of input patterns, so their features are strictly less expressive than those of orientation aware anisotropic filters.

To address this limitation of graph networks we propose *Gauge Equivariant Mesh CNNs* (GEM-CNN), which minimally modify GCNs such that they are able to use anisotropic filters while sharing weights across different positions and respecting the local geometry. One obstacle in sharing anisotropic kernels, which are functions of the angle $\theta_{pq}$ of neighbour $q$ with respect to vertex $p$, over multiple vertices of a mesh is that there is no unique way of selecting a reference neighbour $q_0$, which has the direction $\theta_{pq_0} = 0$. The reference neighbour, and hence the orientation of the neighbours, needs to be chosen arbitrarily. In order to guarantee the equivalence of the features resulting from different choices of orientations, we adapt Gauge Equivariant CNNs (Cohen et al., 2019b) to general meshes. The kernels of our model are thus designed to be *equivariant under gauge transformations*, that is, to guarantee that the responses for different kernel orientations are related by a prespecified transformation law. Such features are identified as geometric objects like scalars, vectors, tensors, etc., depending on the specific choice of transformation law. In order to compare such geometric features at neighbouring vertices, they need to be *parallel transported* along the connecting edge.

In our implementation we first specify the transformation laws of the feature spaces and compute a space of gauge equivariant kernels. Then we pick arbitrary reference orientations at each node, relative to which we compute neighbour orientations and compute the corresponding edge transporters. Given these quantities, we define the forward pass as a message passing step via edge transporters followed by a contraction with the equivariant kernels evaluated at the neighbour orientations. Algorithmically, Gauge Equivariant Mesh CNNs are therefore just GCNs with anisotropic, gauge equivariant kernels and message passing via parallel transporters. Conventional GCNs are covered in this framework for the specific choice of isotropic kernels and trivial edge transporters, given by identity maps.

In Sec. 2, we will give an outline of our method, deferring details to Secs. 3 and 4. In Sec. 3.2, we describe how to compute general geometric quantities, not specific to our method, used for the computation of the convolution. In our experiments in Sec. 6.1, we find that the enhanced expressiveness of Gauge Equivariant Mesh CNNs enables them to outperform conventional GCNs and other prior work in a shape correspondence task.

## 2 Convolutions on Graphs with Geometry

We consider the problem of processing signals on discrete 2-dimensional manifolds, or meshes $M$. Such meshes are described by a set $\mathcal{V}$ of vertices in $\mathbb{R}^3$ together with a set $\mathcal{F}$ of tuples, each consisting of the vertices at the corners of a face. For a mesh to describe a proper manifold, each edge needs to be connected to two faces, and the neighbourhood of each vertex needs to be homeomorphic to a disk. Mesh $M$ induces a graph $\mathcal{G}$ by forgetting the coordinates of the vertices while preserving the edges.

A conventional graph convolution between kernel $K$ and signal $f$, evaluated at a vertex $p$, can be defined by

$$(K \star f)_p = K_{\text{self}} f_p + \sum_{q \in \mathcal{N}_p} K_{\text{neigh}} f_q, \tag{1}$$

where $\mathcal{N}_p$ is the set of neighbours of $p$ in $\mathcal{G}$, and $K_{\text{self}} \in \mathbb{R}^{C_{\text{in}} \times C_{\text{out}}}$ and $K_{\text{neigh}} \in \mathbb{R}^{C_{\text{in}} \times C_{\text{out}}}$ are two linear maps which model a self interaction and the neighbour contribution, respectively. Importantly, graph convolution does not distinguish different neighbours, because each feature vector $f_q$ is multiplied by the same matrix $K_{\text{neigh}}$ and then summed. For this reason we say the kernel is *isotropic*.

Consider the example in figure 1, where on the left and right, the neighbourhood of one vertex $p$, containing neighbours $q \in \mathcal{N}_p$, is visualized. An isotropic kernel would propagate the signal from the neighbours to $p$ in exactly the same way in both neighbourhoods, even though the neighbourhoods are geometrically distinct. For this reason, our method uses direction sensitive (*anisotropic*) kernels instead of isotropic kernels. Anisotropic kernels are inherently more expressive than isotropic ones which is why they are used universally in conventional planar CNNs.

---

**Algorithm 1** Gauge Equivariant Mesh CNN layer

---

**Input:** mesh $M$, input/output feature types $\rho_{\text{in}}, \rho_{\text{out}}$, reference neighbours $(q_0^p \in \mathcal{N}_p)_{p \in M}$.

Compute basis kernels $K_{\text{self}}^i, K_{\text{neigh}}^i(\theta)$          ▷ Sec. 3

Initialise weights $w_{\text{self}}^i$ and $w_{\text{neigh}}^i$.

For each neighbour pair, $p \in M, q \in \mathcal{N}_p$:          ▷ App. A.
     compute neighbor angles $\theta_{pq}$ relative to reference neighbor
     compute parallel transporters $g_{q \to p}$

**Forward**$\big($input features $(f_p)_{p \in M}$, weights $w_{\text{self}}^i, w_{\text{neigh}}^i\big)$:

     $f_p' \leftarrow \sum_i w_{\text{self}}^i K_{\text{self}}^i f_p + \sum_{i, q \in \mathcal{N}_p} w_{\text{neigh}}^i K_{\text{neigh}}^i(\theta_{pq}) \rho_{\text{in}}(g_{q \to p}) f_q$

---

We propose the Gauge Equivariant Mesh Convolution, a minimal modification of graph convolution that allows for anisotropic kernels $K(\theta)$ whose value depends on an orientation $\theta \in [0, 2\pi)$.[1] To define the orientations $\theta_{pq}$ of neighbouring vertices $q \in \mathcal{N}_p$ of $p$, we first map them to the tangent plane $T_p M$ at $p$, as visualized in figure 1. We then pick an *arbitrary* reference neighbour $q_0^p$ to determine a reference orientation[2] $\theta_{pq_0^p} := 0$, marked orange in figure 1. This induces a basis on the tangent plane, which, when expressed in polar coordinates, defines the angles $\theta_{pq}$ of the other neighbours.

As we will motivate in the next section, features in a Gauge Equivariant CNN are coefficients of geometric quantities. For example, a tangent vector at vertex $p$ can be described either geometrically by a 3 dimensional vector orthogonal to the normal at $p$ or by two coefficients in the basis on the tangent plane. In order to perform convolution, geometric features at different vertices need to be linearly combined, for which it is required to first "parallel transport" the features to the same vertex. This is done by applying a matrix $\rho(g_{q \to p}) \in \mathbb{R}^{C_{\text{in}} \times C_{\text{in}}}$ to the coefficients of the feature at $q$, in order to obtain the coefficients of the feature vector transported to $p$, which can be used for the convolution at $p$. The transporter depends on the geometric *type* (group representation) of the feature, denoted by $\rho$ and described in more detail below. Details of how the tangent space is defined, how to compute the map to the tangent space, angles $\theta_{pq}$, and the parallel transporter are given in Appendix A.

In combination, this leads to the GEM-CNN convolution

$$(K \star f)_p = K_{\text{self}} f_p + \sum_{q \in \mathcal{N}_p} K_{\text{neigh}}(\theta_{pq}) \rho(g_{q \to p}) f_q \tag{2}$$

which differs from the conventional graph convolution, defined in Eq. 1 only by the use of an anisotropic kernel and the parallel transport message passing.

We require the outcome of the convolution to be *equivalent* for any choice of reference orientation. This is not the case for any anisotropic kernel but only for those which are *equivariant under changes of reference orientations* (gauge transformations). Equivariance imposes a linear constraint on the kernels. We therefore solve for complete sets of "basis-kernels" $K_{\text{self}}^i$ and $K_{\text{neigh}}^i$ satisfying this constraint and linearly combine them with parameters $w_{\text{self}}^i$ and $w_{\text{neigh}}^i$ such that $K_{\text{self}} = \sum_i w_{\text{self}}^i K_{\text{self}}^i$ and $K_{\text{neigh}} = \sum_i w_{\text{neigh}}^i K_{\text{neigh}}^i$. Details on the computation of basis kernels are given in section 3. The full algorithm for initialisation and forward pass, which is of time and space complexity linear in the number of vertices, for a GEM-CNN layer are listed in algorithm 1. Gradients can be computed by automatic differentiation.

The GEM-CNN is gauge equivariant, but furthermore satisfies two important properties. Firstly, it depends only on the intrinsic shape of the 2D mesh, not on the embedding of the mesh in $\mathbb{R}^3$. Secondly, whenever a map from the mesh to itself exists that preserves distances and orientation, the convolution is equivariant to moving the signal along such transformations. These properties are proven in Appendix D and empirically shown in Appendix F.2.

---

[1] In principle, the kernel could be made dependent on the radial distance of neighboring nodes, by $K_{\text{neigh}}(r, \theta) = F(r) K_{\text{neigh}}(\theta)$, where $F(r)$ is unconstrained and $K_{\text{neigh}}(\theta)$ as presented in this paper. As this dependency did not improve the performance in our empirical evaluation, we omit it.

[2] Mathematically, this corresponds to a choice of *local reference frame* or *gauge*.

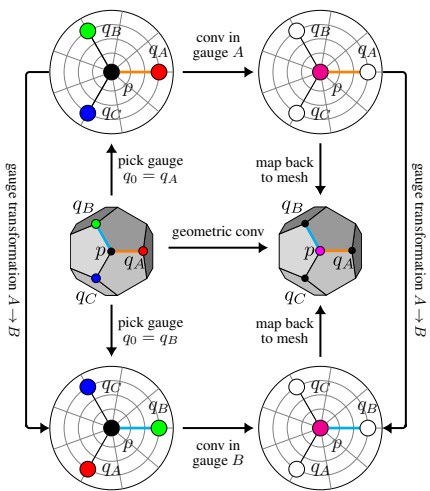 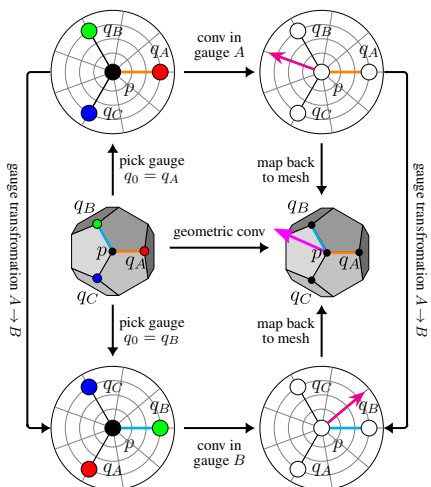

(a) Convolution from scalar to scalar features.  (b) Convolution from scalar to vector features.

Figure 2: Visualization of the Gauge Equivariant Mesh Convolution in two configurations, scalar to scalar and scalar to vector. The convolution operates in a gauge, so that vectors are expressed in coefficients in a basis and neighbours have polar coordinates, but can also be seen as a *geometric convolution*, a gauge-independent map from an input signal on the mesh to a output signal on the mesh. The convolution is equivariant if this geometric convolution does not depend on the intermediate chosen gauge, so if the diagram commutes.

## 3 GAUGE EQUIVARIANCE & GEOMETRIC FEATURES

On a general mesh, the choice of the reference neighbour, or gauge, which defines the orientation of the kernel, can only be made arbitrarily. However, this choice should not arbitrarily affect the outcome of the convolution, as this would impede the generalization between different locations and different meshes. Instead, Gauge Equivariant Mesh CNNs have the property that their output transforms according to a known rule as the gauge changes.

Consider the left hand side of figure 2(a). Given a neighbourhood of vertex $p$, we want to express each neighbour $q$ in terms of its polar coordinates $(r_q, \theta_q)$ on the tangent plane, so that the kernel value at that neighbour $K_{\text{neigh}}(\theta_q)$ is well defined. This requires choosing a basis on the tangent plane, determined by picking a neighbour as reference neighbour (denoted $q_0$), which has the zero angle $\theta_{q_0} = 0$. In the top path, we pick $q_A$ as reference neighbour. Let us call this gauge A, in which neighbours have angles $\theta_q^A$. In the bottom path, we instead pick neighbour $q_B$ as reference point and are in gauge B. We get a different basis for the tangent plane and different angles $\theta_q^B$ for each neighbour. Comparing the two gauges, we see that they are related by a rotation, so that $\theta_q^B = \theta_q^A - \theta_{q_B}^A$. This change of gauge is called a gauge transformation of angle $g := \theta_{q_B}^A$.

In figure 2(a), we illustrate a gauge equivariant convolution that takes input and output features such as gray scale image values on the mesh, which are called scalar features. The top path represents the convolution in gauge A, the bottom path in gauge B. In either case, the convolution can be interpreted as consisting of three steps. First, for each vertex $p$, the value of the scalar features on the mesh at each neighbouring vertex $q$, represented by colors, is mapped to the tangent plane at $p$ at angle $\theta_q$ defined by the gauge. Subsequently, the convolutional kernel sums for each neighbour $q$, the product of the feature at $q$ and kernel $K(\theta_q)$. Finally the output is mapped back to the mesh. These three steps can be composed into a single step, which we could call a *geometric convolution*, mapping from input features on the mesh to output features on the mesh. The convolution is *gauge equivariant* if this geometric convolution does not depend on the gauge we pick in the interim, so in figure 2(a), if the convolution in the top path in gauge A has same result the convolution in the bottom path in gauge B, making the diagram commute. In this case, however, we see that the convolution output needs to be the same in both gauges, for the convolution to be equivariant. Hence, we must have that $K(\theta_q) = K(\theta_q - g)$, as the orientations of the neighbours differ by some angle $g$, and the kernel must be isotropic.

As we aim to design an anisotropic convolution, the output feature of the convolution at $p$ can, instead of a scalar, be two numbers $v \in \mathbb{R}^2$, which can be interpreted as coefficients of a tangent feature

vector in the tangent space at $p$, visualized in figure 2(b). As shown on the right hand side, different gauges induce a different basis of the tangent plane, so that the *same tangent vector* (shown on the middle right on the mesh), is represented by *different coefficients* in the gauge (shown on the top and bottom on the right). This gauge equivariant convolution must be anisotropic: going from the top row to the bottom row, if we change orientations of the neighbours by $-g$, the coefficients of the output vector $v \in \mathbb{R}^2$ of the kernel must be also rotated by $-g$. This is written as $R(-g)v$, where $R(-g) \in \mathbb{R}^{2 \times 2}$ is the matrix that rotates by angle $-g$.

Vectors and scalars are not the only type of geometric features that can be inputs and outputs of a GEM-CNN layer. In general, the coefficients of a geometric feature of $C$ dimensions changes by an invertible linear transformation $\rho(-g) \in \mathbb{R}^{C \times C}$ if the gauge is rotated by angle $g$. The map $\rho : [0, 2\pi) \to \mathbb{R}^{C \times C}$ is called the *type* of the geometric quantity and is formally known as a group representation of the planar rotation group $\mathrm{SO}(2)$. Group representations have the property that $\rho(g + h) = \rho(g)\rho(h)$ (they are group homomorphisms), which implies in particular that $\rho(0) = \mathbb{1}$ and $\rho(-g) = \rho(g)^{-1}$. For more background on group representation theory, we refer the reader to (Serre, 1977) and, specifically in the context of equivariant deep learning, to (Lang & Weiler, 2020). From the theory of group representations, we know that any feature type can be composed from "irreducible representations" (irreps). For $\mathrm{SO}(2)$, these are the one dimensional invariant scalar representation $\rho_0$ and for all $n \in \mathbb{N}_{>0}$, a two dimensional representation $\rho_n$,

$$\rho_0(g) = 1, \quad \rho_n(g) = \begin{pmatrix} \cos ng & -\sin ng \\ \sin ng & \cos ng \end{pmatrix}.$$

where we write, for example, $\rho = \rho_0 \oplus \rho_1 \oplus \rho_1$ to denote that representation $\rho(g)$ is the direct sum (i.e. block-diagonal stacking) of the matrices $\rho_0(g), \rho_1(g), \rho_1(g)$. Scalars and tangent vector features correspond to $\rho_0$ and $\rho_1$ respectively and we have $R(g) = \rho_1(g)$.

The type of the feature at each layer in the network can thus be fully specified (up to a change of basis) by the number of copies of each irrep. Similar to the dimensionality in a conventional CNN, the choice of type is a hyperparameter that can be freely chosen to optimize performance.

## 3.1 KERNEL CONSTRAINT

Given an input type $\rho_{\text{in}}$ and output type $\rho_{\text{out}}$ of dimensions $C_{\text{in}}$ and $C_{\text{out}}$, the kernels are $K_{\text{self}} \in \mathbb{R}^{C_{\text{out}} \times C_{\text{in}}}$ and $K_{\text{neigh}} : [0, 2\pi) \to \mathbb{R}^{C_{\text{out}} \times C_{\text{in}}}$. However, not all such kernels are equivariant. Consider again examples figure 2(a) and figure 2(b). If we map from a scalar to a scalar, we get that $K_{\text{neigh}}(\theta - g) = K_{\text{neigh}}(\theta)$ for all angles $\theta, g$ and the convolution is isotropic. If we map from a scalar to a vector, we get that rotating the angles $\theta_q$ results in the same tangent vector as rotating the output vector coefficients, so that $K_{\text{neigh}}(\theta - g) = R(-g)K_{\text{neigh}}(\theta)$.

In general, as derived by Cohen et al. (2019b) and in appendix B, the kernels must satisfy for any gauge transformation $g \in [0, 2\pi)$ and angle $\theta \in [0, 2\pi)$, that

$$K_{\text{neigh}}(\theta - g) = \rho_{\text{out}}(-g)K_{\text{neigh}}(\theta)\rho_{\text{in}}(g), \quad (3)$$
$$K_{\text{self}} = \rho_{\text{out}}(-g) \, K_{\text{self}} \, \rho_{\text{in}}(g). \quad (4)$$

The kernel can be seen as consisting of multiple blocks, where each block takes as input one irrep and outputs one irrep. For example if $\rho_{\text{in}}$ would be of type $\rho_0 \oplus \rho_1 \oplus \rho_1$ and $\rho_{\text{out}}$ of type $\rho_1 \oplus \rho_3$, we have $4 \times 5$ matrix

$$K_{\text{neigh}}(\theta) = \begin{pmatrix} K_{10}(\theta) & K_{11}(\theta) & K_{11}(\theta) \\ K_{30}(\theta) & K_{31}(\theta) & K_{31}(\theta) \end{pmatrix}$$

| $\rho_{\text{in}} \to \rho_{\text{out}}$ | linearly independent solutions for $K_{\text{neigh}}(\theta)$ |
|---|---|
| $\rho_0 \to \rho_0$ | $(1)$ |
| $\rho_n \to \rho_0$ | $(\cos n\theta \ \sin n\theta), (\sin n\theta \ -\cos n\theta)$ |
| $\rho_0 \to \rho_m$ | $\begin{pmatrix} \cos m\theta \\ \sin m\theta \end{pmatrix}, \begin{pmatrix} \sin m\theta \\ -\cos m\theta \end{pmatrix}$ |
| $\rho_n \to \rho_m$ | $\begin{pmatrix} c_- & -s_- \\ s_- & c_- \end{pmatrix}, \begin{pmatrix} s_- & c_- \\ -c_- & s_- \end{pmatrix}, \begin{pmatrix} c_+ & s_+ \\ s_+ & -c_+ \end{pmatrix}, \begin{pmatrix} -s_+ & c_+ \\ c_+ & s_+ \end{pmatrix}$ |

| $\rho_{\text{in}} \to \rho_{\text{out}}$ | linearly independent solutions for $K_{\text{self}}$ |
|---|---|
| $\rho_0 \to \rho_0$ | $(1)$ |
| $\rho_n \to \rho_n$ | $\begin{pmatrix} 1 & 0 \\ 0 & 1 \end{pmatrix}, \begin{pmatrix} 0 & 1 \\ -1 & 0 \end{pmatrix}$ |

Table 1: Solutions to the angular kernel constraint for kernels that map from $\rho_n$ to $\rho_m$. We denote $c_{\pm} = \cos((m \pm n)\theta)$ and $s_{\pm} = \sin((m \pm n)\theta)$.

where e.g. $K_{31}(\theta) \in \mathbb{R}^{2 \times 2}$ is a kernel that takes as input irrep $\rho_1$ and as output irrep $\rho_3$ and needs to satisfy Eq. 3. As derived by Weiler & Cesa (2019) and in Appendix C, the kernels $K_{\text{neigh}}(\theta)$ and $K_{\text{self}}$ mapping from irrep $\rho_n$ to irrep $\rho_m$ can be written as a linear combination of the basis kernels listed in Table 1. The table shows that equivariance requires the self-interaction to only map from one irrep to the same irrep. Hence, we have $K_{\text{self}} = \begin{pmatrix} 0 & K_{11} & K_{11} \\ 0 & 0 & 0 \end{pmatrix} \in \mathbb{R}^{4 \times 3}$.

All basis-kernels of all pairs of input irreps and output irreps can be linearly combined to form an arbitrary equivariant kernel from feature of type $\rho_{\text{in}}$ to $\rho_{\text{out}}$. In the above example, we have $2 \times 2 + 4 \times 4 = 20$ basis kernels for $K_{\text{neigh}}$ and 4 basis kernels for $K_{\text{self}}$. The layer thus has 24 parameters. As proven in (Weiler & Cesa, 2019) and (Lang & Weiler, 2020), this parameterization of the equivariant kernel space is *complete*, that is, more general equivariant kernels do not exist.

## 3.2 GEOMETRY AND PARALLEL TRANSPORT

In order to implement gauge equivariant mesh CNNs, we need to make the abstract notion of tangent spaces, gauges and transporters concrete.

As the mesh is embedded in $\mathbb{R}^3$, a natural definition of the tangent spaces $T_p M$ is as two dimensional subspaces that are orthogonal to the normal vector at $p$. We follow the common definition of normal vectors at mesh vertices as the area weighted average of the adjacent faces' normals. The Riemannian logarithm map $\log_p : \mathcal{N}_p \to T_p M$ represents the one-ring neighborhood of each point $p$ on their tangent spaces as visualized in figure 1. Specifically, neighbors $q \in \mathcal{N}_p$ are mapped to $\log_p(q) \in T_p M$ by first projecting them to $T_p M$ and then rescaling the projection such that the norm is preserved, i.e. $|\log_p(q)| = |q - p|$; see Eq. 6. A choice of reference neighbor $q_p \in \mathcal{N}$ uniquely determines a right handed, orthonormal reference frame $(e_{p,1}, e_{p,2})$ of $T_p M$ by setting $e_{p,1} := \log_p(q_0)/|\log_p(q_0)|$ and $e_{p,2} := n \times e_{p,1}$. The polar angle $\theta_{pq}$ of any neighbor $q \in \mathcal{N}$ relative to the first frame axis is then given by $\theta_{pq} := \text{atan2}\left(e_{p,2}^\top \log_p(q), e_{p,1}^\top \log_p(q)\right)$.

Given the reference frame $(e_{p,1}, e_{p,2})$, a 2-tuple of coefficients $(v_1, v_2) \in \mathbb{R}^2$ specifies an (embedded) tangent vector $v_1 e_{p,1} + v_2 e_{p,2} \in T_p M \subset \mathbb{R}^3$. This assignment is formally given by the *gauge map* $E_p : \mathbb{R}^2 \to T_p M \subset \mathbb{R}^3$ which is a vector space isomorphism. In our case, it can be identified with the matrix

$$E_p = \left[ \begin{array}{cc} | & | \\ e_{p,1} & e_{p,2} \\ | & | \end{array} \right] \in \mathbb{R}^{3 \times 2}. \tag{5}$$

Feature vectors $f_p$ and $f_q$ at neighboring (or any other) vertices $p \in M$ and $q \in \mathcal{N}_p \subseteq M$ live in different vector spaces and are expressed relative to independent gauges, which makes it invalid to sum them directly. Instead, they have to be parallel transported along the mesh edge that connects the two vertices. As explained above, this transport is given by group elements $g_{q \to p} \in [0, 2\pi)$, which determine the transformation of tangent vector *coefficients* as $v_q \mapsto R(g_{q \to p})v_q \in \mathbb{R}^2$ and, analogously, for feature vector coefficients as $f_q \mapsto \rho(g_{q \to p})f_q$. Figure 4 in the appendix visualizes the definition of edge transporters for flat spaces and meshes. On a flat space, tangent vectors are transported by keeping them parallel in the usual sense on Euclidean spaces. However, if the source and target frame orientations disagree, the vector coefficients relative to the source frame need to be transformed to the target frame. This coordinate transformation from polar angles $\varphi_q$ of $v$ to $\varphi_p$ of $R(g_{q \to p})v$ defines the transporter $g_{q \to p} = \varphi_p - \varphi_q$. On meshes, the source and target tangent spaces $T_q M$ and $T_p M$ are not longer parallel. It is therefore additionally necessary to rotate the source tangent space and its vectors parallel to the target space, before transforming between the frames. Since transporters effectively make up for differences in the source and target frames, the parallel transporters transform under gauge transformations $g_p$ and $g_q$ according to $g_{q \to p} \mapsto g_p + g_{q \to p} - g_q$. Note that this transformation law cancels with the transformation law of the coefficients at $q$ and lets the transported coefficients transform according to gauge transformations at $p$. It is therefore valid to sum vectors and features that are parallel transported into the same gauge at $p$.

A more detailed discussion of the concepts presented in this section can be found in Appendix A.

## 4 NON-LINEARITY

Besides convolutional layers, the GEM-CNN contains non-linear layers, which also need to be gauge equivariant, for the entire network to be gauge equivariant. The coefficients of features built out of irreducible representaions, as described in section 3, do not commute with point-wise non-linearities (Worrall et al., 2017; Thomas et al., 2018; Weiler et al., 2018a; Kondor et al., 2018). Norm non-linearities and gated non-linearities (Weiler & Cesa, 2019) can be used with such features, but generally perform worse in practice compared to point-wise non-linearities (Weiler & Cesa,

2019). Hence, we propose the *RegularNonlinearity*, which uses point-wise non-linearities and is approximately gauge equivariant.

This non-linearity is built on Fourier transformations. Consider a continuous periodic signal, on which we perform a band-limited Fourier transform with band limit $b$, obtaining $2b + 1$ Fourier coefficients. If this continuous signal is shifted by an arbitrary angle $g$, then the corresponding Fourier components transform with linear transformation $\rho_{0:b}(-g)$, for $2b + 1$ dimensional representation $\rho_{0:b} := \rho_0 \oplus \rho_1 \oplus ... \oplus \rho_b$.

It would be exactly equivariant to take a feature of type $\rho_{0:b}$, take a continuous inverse Fourier transform to a continuous periodic signal, then apply a point-wise non-linearity to that signal, and take the continuous Fourier transform, to recover a feature of type $\rho_{0:b}$. However, for implementation, we use $N$ intermediate samples and the discrete Fourier transform. This is exactly gauge equivariant for gauge transformation of angles multiple of $2\pi/N$, but only approximately equivariant for other angles. In App. G we prove that as $N \to \infty$, the non-linearity is exactly gauge equivariant.

The run-time cost per vertex of the (inverse) Fourier transform implemented as a simple linear transformation is $\mathcal{O}(bN)$, which is what we use in our experiments. The pointwise non-linearity scales linearly with $N$, so the complexity of the RegularNonLineariy is also $\mathcal{O}(bN)$. However, one can also use a fast Fourier transform, achieving a complexity of $\mathcal{O}(N \log N)$. Concrete memory and run-time cost of varying $N$ are shown in appendix F.1.

## 5    RELATED WORK

The irregular structure of meshes leads to a variety of approaches to define convolutions. Closely related to our method are graph based methods which are often based on variations of graph convolutional networks (Kipf & Welling, 2017; Defferrard et al., 2016). GCNs have been applied on spherical meshes (Perraudin et al., 2019) and cortical surfaces (Cucurull et al., 2018; Zhao et al., 2019a). Verma et al. (2018) augment GCNs with anisotropic kernels which are dynamically computed via an attention mechanism over graph neighbours.

Instead of operating on the graph underlying a mesh, several approaches leverage its geometry by treating it as a discrete manifold. Convolution kernels can then be defined in geodesic polar coordinates which corresponds to a projection of kernels from the tangent space to the mesh via the exponential map. This allows for kernels that are larger than the immediate graph neighbourhood and message passing over faces but does not resolve the issue of ambiguous kernel orientation. Masci et al. (2015); Monti et al. (2016) and Sun et al. (2018) address this issue by restricting the network to orientation invariant features which are computed by applying anisotropic kernels in several orientations and pooling over the resulting responses. The models proposed in (Boscaini et al., 2016) and (Schonsheck et al., 2018) are explicitly gauge dependent with preferred orientations chosen via the principal curvature direction and the parallel transport of kernels, respectively. Poulenard & Ovsjanikov (2018) proposed a non-trivially gauge equivariant network based on geodesic convolutions, however, the model parallel transports only partial information of the feature vectors, corresponding to certain kernel orientations. In concurrent work, Wiersma et al. (2020) also define convolutions on surfaces equivariantly to the orientation of the kernel, but differ in that they use norm non-linearities instead of regular ones and that they apply the convolution along longer geodesics, which adds complexity to the geometric pre-computation - as partial differential equations need to be solved, but may result in less susceptibility to the particular discretisation of the manifold.

Another class of approaches defines spectral convolutions on meshes. However, as argued in (Bronstein et al., 2017), the Fourier spectrum of a mesh depends heavily on its geometry, which makes such methods instable under deformations and impedes the generalization between different meshes. Spectral convolutions further correspond to isotropic kernels. Kostrikov et al. (2018) overcomes isotropy of the Laplacian by decomposing it into two applications of the first-order Dirac operator.

A construction based on toric covering maps of topologically spherical meshes was presented in (Maron et al., 2017). An entirely different approach to mesh convolutions is to apply a linear map to a spiral of neighbours (Bouritsas et al., 2019; Gong et al., 2019), which works well only for meshes with a similar graph structure.

The above-mentioned methods operate on the intrinsic, 2-dimensional geometry of the mesh. A popular alternative for embedded meshes is to define convolutions in the embedding space $\mathbb{R}^3$. This can for instance be done by voxelizing space and representing the mesh in terms of an occupancy grid (Wu et al., 2015; Tchapmi et al., 2017; Hanocka et al., 2018). A downside of this approach are the high memory and compute requirements of voxel representations. If the grid occupancy is low, this can partly be addressed by resorting to an inhomogeneous grid density (Riegler et al., 2017). Instead of voxelizing space, one may interpret the set of mesh vertices as a point cloud and run a convolution on those (Qi et al., 2017a;b). Point cloud based methods can be made equivariant w.r.t. the isometries of $\mathbb{R}^3$ (Zhao et al., 2019b; Thomas et al., 2018), which implies in particular the isometry equivariance on the embedded mesh. In general, geodesic distances within the manifold differ usually substantially from the distances in the embedding space. Which approach is more suitable depends on the particular application.

On flat Euclidean spaces our method corresponds to Steerable CNNs (Cohen & Welling, 2017; Weiler et al., 2018a; Weiler & Cesa, 2019; Cohen et al., 2019a; Lang & Weiler, 2020). As our model, these networks process geometric feature fields of types $\rho$ and are equivariant under gauge transformations, however, due to the flat geometry, the parallel transporters become trivial. Regular nonlinearities are on flat spaces used in group convolutional networks (Cohen & Welling, 2016; Weiler et al., 2018b; Hoogeboom et al., 2018; Bekkers et al., 2018; Winkels & Cohen, 2018; Worrall & Brostow, 2018; Worrall & Welling, 2019; Sosnovik et al., 2020).

# 6 EXPERIMENTS

## 6.1 EMBEDDED MNIST

We first investigate how Gauge Equivariant Mesh CNNs perform on, and generalize between, different mesh geometries. For this purpose we conduct simple MNIST digit classification experiments on embedded rectangular meshes of $28 \times 28$ vertices. As a baseline geometry we consider a flat mesh as visualized in figure 5(a). A second type of geometry is defined as different *isometric* embeddings of the flat mesh, see figure 5(b). Note that this implies that the *intrinsic* geometry of these isometrically embedded meshes is indistinguishable from that of the flat mesh. To generate geometries which are intrinsically curved, we add random normal displacements to the flat mesh. We control the amount of curvature by smoothing the resulting displacement fields with Gaussian kernels of different widths $\sigma$ and define the roughness of the resulting mesh as $3 - \sigma$. Figures 5(c)-5(h) show the results for roughnesses of 0.5, 1, 1.5, 2, 2.25 and 2.5. For each of the considered settings we generate 32 different train and 32 test geometries.

To test the performance on, and generalization between, different geometries, we train equivalent GEM-CNN models on a flat mesh and meshes with a roughness of 1, 1.5, 2, 2.25 and 2.5. Each model is tested individually on each of the considered test geometries, which are the flat mesh, isometric embeddings and curved embeddings with a roughness of 0.5, 1, 1.25, 1.5, 1.75, 2, 2.25 and 2.5. Figure 3 shows the test errors of the GEM-CNNs on the different train geometries (different curves) for all test geometries (shown on the x-axis). Since our model is purely defined in terms of the intrinsic geometry of a mesh, it is expected to be insensitive to isometric changes in the embeddings. This is empirically confirmed by the fact that the test performances on flat and isometric embeddings are exactly equal. As expected, the test error increases for most models with the surface roughness. Models trained on more rough surfaces are hereby more robust to deformations. The models generalize well from a rough training to smooth test geometry up to a training roughness of 1.5. Beyond that point, the test performances on smooth meshes degrades up to the point of random guessing at a training roughness of 2.5.

As a baseline, we build an *isotropic* graph CNN with the same network topology and number of parameters ($\approx 163k$). This model is insensitive to the mesh geometry and therefore performs exactly equal on all surfaces. While this enhances its robustness on very rough meshes, its test error of $19.80 \pm 3.43\%$ is an extremely bad result on MNIST. In contrast, the use of anisotropic filters of GEM-CNN allows it to reach a test error of only $0.60 \pm 0.05\%$ on the flat geometry. It is therefore competitive with conventional CNNs on pixel grids, which apply anisotropic kernels as well. More details on the datasets, models and further experimental setup are given in appendix E.1.

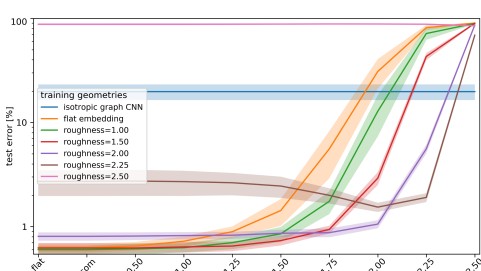

| Model | Features | Accuracy (%) |
|---|---|---|
| ACNN (Boscaini et al., 2016) | SHOT | 62.4 |
| Geodesic CNN (Masci et al., 2015) | SHOT | 65.4 |
| MoNet (Monti et al., 2016) | SHOT | 73.8 |
| FeaStNet (Verma et al., 2018) | XYZ | 98.7 |
| ZerNet (Sun et al., 2018) | XYZ | 96.9 |
| SpiralNet++ (Gong et al., 2019) | XYZ | 99.8 |
| Graph CNN | XYZ | 1.40±0.5 |
| Graph CNN | SHOT | 23.80±8 |
| Non-equiv. CNN (SHOT frames) | XYZ | 73.00±4.0 |
| Non-equiv. CNN (SHOT frames) | SHOT | 75.11±2.4 |
| GEM-CNN | XYZ | 99.73±0.04 |
| GEM-CNN (broken symmetry) | XYZ | **99.89**±0.02 |

Figure 3: Test errors for MNIST digit classification on embedded meshes. Different lines denote train geometries, x-axis shows test geometries. Regions are standard errors of the means over 6 runs.

Table 2: Results of FAUST shape correspondence. Statistics are means and standard errors of the mean of over three runs. All cited results are from their respective papers.

## 6.2 SHAPE CORRESPONDENCE

As a second experiment, we perform non-rigid shape correspondence on the FAUST dataset (Bogo et al., 2014), following Masci et al. (2015) [3] . The data consists of 100 meshes of human bodies in various positions, split into 80 train and 20 test meshes. The vertices are registered, such that vertices on the same position on the body, such as the tip of the left thumb, have the same identifier on all meshes. All meshes have 6890 vertices, making this a 6890-class segmentation problem.

The architecture transforms the vertices' $XYZ$ coordinates (of type $3\rho_0$), via 6 convolutional layers to features $64\rho_0$, with intermediate features $16(\rho_0 \oplus \rho_1 \oplus \rho_2)$, with residual connections and the RegularNonlinearity with $N = 5$ samples. Afterwards, we use two $1\times1$ convolutions with ReLU to map first to 256 and then 6980 channels, after which a softmax predicts the registration probabilities. The $1\times1$ convolutions use a dropout of 50% and 1E-4 weight decay. The network is trained with a cross entropy loss with an initial learning rate of 0.01, which is halved when training loss reaches a plateau.

As all meshes in the FAUST dataset share the same topology, breaking the gauge equivariance in higher layers can actually be beneficial. As shown in (Weiler & Cesa, 2019), symmetry can be broken by treating non-invariant features as invariant features as input to the final $1\times1$ convolution.

As baselines, we compare to various models, some of which use more complicated pipelines, such as (1) the computation of geodesics over the mesh, which requires solving partial differential equations, (2) pooling, which requires finding a uniform sub-selection of vertices, (3) the pre-computation of SHOT features which locally describe the geometry (Tombari et al., 2010), or (4) post-processing refinement of the predictions. The GEM-CNN requires none of these additional steps. In addition, we compare to SpiralNet++ (Gong et al., 2019), which requires all inputs to be similarly meshed. Finally, we compare to an isotropic version of the GEM-CNN, which reduces to a conventional graph CNN, as well as a non-gauge-equivariant CNN based on SHOT frames. The results in table 2 show that the GEM-CNN outperforms prior works and a non-gauge-equivariant CNN, that isotropic graph CNNs are unable to solve the task and that for this data set breaking gauge symmetry in the final layers of the network is beneficial. More experimental details are given in appendix E.2.

## 7 CONCLUSIONS

Convolutions on meshes are commonly performed as a convolution on their underlying graph, by forgetting geometry, such as orientation of neighbouring vertices. In this paper we propose Gauge Equivariant Mesh CNNs, which endow Graph Convolutional Networks on meshes with anisotropic kernels and parallel transport. Hence, they are sensitive to the mesh geometry, and result in equivalent outputs regardless of the arbitrary choice of kernel orientation.

We demonstrate that the inference of GEM-CNNs is invariant under isometric deformations of meshes and generalizes well over a range of non-isometric deformations. On the FAUST shape correspondence task, we show that Gauge equivariance, combined with symmetry breaking in the final layer, leads to state of the art performance.

---

[3]These experiments were executed on QUVA machines.

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
