# OpenReview forum: "Gauge Equivariant Mesh CNNs: Anisotropic convolutions on geometric graphs"
_ICLR.cc/2021/Conference — ICLR 2021 Spotlight_

### Official Review · AnonReviewer4 · 2020-10-28
**Elegant way to incorporate gauge symmetry on meshes; RegularNonlinearity addresses an important issue; prefer more interesting experiments**

**Rating:** 7
**Confidence:** 4

**Review:**

Although a mesh embedded in 3D space may be treated as a graph, a graph convolution network uses the same weights for each neighbor and is thus permutation invariant, which is the incorrect inductive bias for a mesh: the neighbors of a node are spatially related and may not be arbitrarily permuted.  CNNs, GCNs, and G-CNNs demonstrate the value of a weight sharing scheme which correctly reflects the symmetry of the underlying space of the data.  The authors argue convincingly that for a signal on a mesh, the appropriate bias is symmetry to local change-of-gauge.  In short, the weights should depend on the relative orientation of a node’s neighbors.  They design a network GEM-CNN which is equivariant to change of gauge.  The design is similar to a GCN but incorporates parallel transport to account for underlying geometry and uses kernels similar to those of $SO(2)$-equivariant $E(2)$-CNN (Weiler & Cesa 2019).  The experiments show the network is able to adapt to different mesh geometries and obtain very high accuracy in the shape correspondence task.

I suggest accepting this paper.  The architecture is an elegant way to incorporate gauge symmetry on meshes and RegularNonlinearity addresses an important issue for equivariant neural nets.  Though I would prefer more interesting experiments, they are sufficient to validate the design.

**Specific Strength, Weaknesses, Points, and Questions:**
- The symmetry of a graph convolutional network can be broken by including spatial coordinates as features.  If $x_p$ and $x_q$ are inputs, then a function $F(f_q,x_q,x_p)$ can process data in an orientation aware way even though $F$ is isotropic.  How would this compare to the proposed method in the paper?
-Page 2: One strength of the paper is the argument for why gauge symmetry is necessary in the first place.  Features defined on a mesh may be vector-valued and defined with respect to a frame of reference at each point on the mesh.  However, since the geometry is curved, there is no consistent way to assign a frame of reference.  Thus an arbitrary choice must be made when recording data.  Since this choice is arbitrary, the output of the network should clearly be independent of it.  Gauge equivariance encodes this symmetry.
-The argument for encoding the geometry of the mesh is reasonable.  But then why only parameterize $K$ by $\theta_q$; why not also include the distance $r_q$?
-Page 6: **RegularNonlinearity is an important contribution** of the paper.  The authors are correct that non-linearities have been a bottleneck for using equivariant neural networks with representations other than the regular representation.  Transforming to sample space (i.e. embedding in the regular representation) to apply a pointwise non-linearity and then transforming back is a nice idea for addressing this.  Furthermore, Theorem E.1 provides a nice theoretical analysis of the asymptotic error.  It would be nice to include practical non-asymptotic error bounds as well.
-The experiments are okay, but are a weaker part of the paper.  The argument for a geometry-aware NN is that it processes signals on the geometry better.  It is not clear that embedding MNIST on a mesh illustrates data which is best understood in terms of the underlying mesh.  Far better would be to develop a signal natively on the mesh, for example by solving a PDE directly on the mesh.  Arguably the FAUST shape correspondence data addresses this issue better since the signal is inherently linked to the geometry.
-Both experiments also treat only scalar data of type $\rho_0$.  While it is plausible and reasonable to model such data using vector features of type $\rho_i$ in the hidden layers, the argument for the necessity of gauge equivariance would be even stronger if the input and/or output signal was itself vector valued, for example a velocity or gradient on the mesh.
-Changing roughness to the embedded MNIST distorts the signal in the geometry of the manifold (changing distances and angles), so why should we expect generalization across different roughnesses?
-Page 7: I don’t understand the argument for the value of symmetry breaking.  The gauge equivariant network can be orientation aware by encoding $\rho_1$ features.  Why is it desirable to be dependent on arbitrary gauge choices?  What breaks down about the original argument for incorporating equivariance in this case?  Is it possible the improvement is due to a different trade off between bias and expressivity at lower layers?
-Page 7, Para 4: The paper argues other high performing methods in shape correspondence use complicated pipelines.  It is not clear to me (probably from lack of familiarity) which is most complicated.  It seems both this method and the other method contain different complexities and subtleties.

**Minor Points:**
- All of the citations in the paper use \citet, but it would be more readable to use \citep.
- Page 3: Should not $\rho(g_{q\to p})$ be invertible in $\mathbb{R}^{c_{in} \times c_{in}}$?
- Page 5: The notation $k \rho_l$ is non-standard, compared to $\rho_l^k$, but it is more readable.
- Page 13, $K_{neigh} \theta_{pq}-g)$ is missing a parenthesis
- Page 13, “which is true for any features, if”.  if could be if and only if, correct?

**Update From Author Reply**
I am grateful for the author's replies, edits, and additional evaluation, all performed within limited time. This helps me feel confident in my accept (7) recommendation. My reason for not giving a higher score remains the limited experiments (which is likely not something to be addressed in two weeks), but even so I think the work is quite worthy of being accepted. The methods are a significant contribution and the experiments are sufficient to demonstrate they work.

---

> ### Author Response · Authors · 2020-11-20
> **Thank you for the review and the helpful suggestions**
>
> We thank the reviewer for their kind words and thorough review. We’d like to address the following points:
> * The reviewer proposes an alternative method based on a function F(f_q, x_q, x_p). This is a great idea and applied widely in practice. We refer to such methods as point-cloud methods and have added a paragraph in the related works section in the revised version. If one desires such a method to be equivariant to global transformations of the vector coordinates, one requires constraints on such a function F and arrives at a method like Tensor Field Networks.
> * “why not include the distance r_q?” Please see our response to all reviewers jointly.
> * “Both experiments also treat only scalar data of type rho_0”. We use datasets with scalar inputs and outputs, because these were easily available. We agree with the reviewer that on problems with higher-order inputs and outputs our method is indeed even more applicable.
> * “Changing roughness to the embedded MNIST distorts …”. The idea of the experiment is to show that under isometric transformations of the coordinates, the output of the method is invariant and, as the reviewer correctly notes, when the mesh is stretched, the output is no longer invariant. This property demonstrates the fact that our method is indeed a method that depends only on the intrinsic 2D shape of the mesh.
> * “I don’t understand the argument for the value of symmetry breaking.” We agree that it is somewhat counter-intuitive that the symmetry breaking aids performance. We suspect that this is because in FAUST all shapes are meshed uniformly, that there is some additional information encoded in the ordering of the edges in each face, and hence in the choice of reference neighbour, which is based on this ordering. In such cases it can be beneficial to be equivariant in lower layers of the network, and break equivariance in higher layers.
> * “The paper argues other high performing methods in shape correspondence use complicated pipelines.” Please see our response to all reviewers jointly about this important design choice.
> * We have changed \citet to \citep and k rho to rho^k, which indeed looks better. Thanks!
> * The rho matrices are indeed always invertible, as they are group representations. We’ve added this to section 3.
> * “which is true for any features, if” Thank you for spotting this, indeed the stronger statement of “if and only if” is true. We’ve changed this in the revised version.

---

> > ### Comment · AnonReviewer4 · 2020-11-21
> > **clarification on minor point 2**
> >
> > Just a quick clarification.  Minor point 2 is that on Page 3, Para 3, you write $\rho(g_{q\to p}) \in \mathbb{R}^{C_{out}\times C_{in}}$ but I think this matrix should be square.  i.e.  $\rho(g_{q\to p}) \in \mathbb{R}^{C_{in}\times C_{in}}$.

---

> > > ### Author Response · Authors · 2020-11-22
> > > **Thank you, will fix**
> > >
> > > Thank you for the clarification and spotting the typo. This should be $\mathbb{R}^{C_{in} \times C_{in}}$, as we transport an input feature along the geodesic from $p$ to $q$. We'll fix this in the final revision.

---

> ### Author Response · Authors · 2020-11-25
> **Implemented empirical evaluation of equivariance on non-linearity**
>
> Following your great suggestion, we have added in appendix F.2 an analysis of the practical non-asymptotic equivariance error caused by the non-linearity.

---

### Official Review · AnonReviewer1 · 2020-10-28
**A novel and interesting method with weak experiments**

**Rating:** 7
**Confidence:** 4

**Review:**

The work presents a novel message passing GNN operator for meshes that is equivariant under gauge transformations. It achieves that by parallel transporting features along edges and spanning a space of gauge equivariant kernels. Further, a DFT-based non-linearity is proposed, which preserves the equivariance in the limit of sampling density. The method is evaluated on an MNIST toy experiment and the Faust shape correspondence task.

Strengths:
- The method is very elegant and novel and seems to be one of the most sophisticated mesh GNN operator so far.
- At the same time it looks like that the operator stays reasonably efficient, still keeping linear time complexity in number of edges. I would welcome, though, if computational efficiency would be analyzed in the work.
- It bridges the more theoretical equivariant convolutions with graph neural networks for mesh processing, which are more commonly used in practice.
- An equivariant non-linearity based on the discrete Fourier transform is presented.
- The work seems to be technically and mathematically sound.
- The paper is well written.
- The figures complement the text well and are helpful for understanding.

Weaknesses:
- Neither the MNIST nor the FAUST experiment verify the gauge equivariance. I think a toy experiment verifying it would be necessary, especially since equivariance of the non-linearity is only approximated (maybe by showing appropriate feature histograms after the non-linearity for changing reference points).
- In addition to a missing verification experiment, it is also hard to follow the theoretical reasoning that the whole approach is equivariant. The line of reasoning needs to be gathered from the main text, appendix, and referenced related work. I think with more concrete theorems, clarity could be improved.
- The Faust shape correspondence task is not sufficient to evaluate a novel mesh operator. From personal experience I know that at least SpiralNet++ (anisotropic, intrinsic, fixed topology) and SplineCNN (anisotropic, extrinsic, arbitrary/varying topology) can be tuned to reach near perfect accuracy on this task. Such small differences in performance can come down to architecture design and might not come from more principled differences in the method. Therefore, I would highly welcome one additional comparison on a different mesh task.
- Since the experiments are scarse, I wonder if the operator is hard to apply to larger tasks or if the kernel restrictions weaken the approach on other tasks.


Questions and comments:
- As usual in this area, the kernels are restricted in trade for equivariance. I wonder how expressive the kernels still are. Is there a way to compare the expressiveness (visually or quantitatively) to lets say an MLP kernel function mapping polar or Cartesian coordinates to a full C x C matrix? Is it the/a minimal restriction that ensures equivariance or is it more restrictive?
- Would it be possible to come up with a space of two-dimensional kernels K_neigh (dependent on the full polar coordinates, including r) while keeping the gauge equivariance?
- How efficient is the non-linearity based on DFT? I would be interested in a execution time breakdown for the whole method, showing the bottlenecks.
- In the MNIST experiment, pooling is applied (appendix D.1). How does this interfere with the equivariance property?

Minor/Typo:
- Appendix page 13, equation above eq 8, parenthesis missing

Overall, I like the paper, it is nice to read and the method is interesting, building on mathematically elegant concepts.  I actually would like to give an accept score. However, in my opinion there are experiments missing: (1, crucial) verifying the gauge equivariance and (2) an additional comparison on a more complex task. Without those experiments (especially 1), I see the paper in a borderline state.

---

> ### Author Response · Authors · 2020-11-20
> **Thank you for the review and the helpful suggestions**
>
> We thank the reviewer for their appreciation of the strengths of the paper and their constructive comments. We’d like to respond to the following points:
> * We agree with the reviewer that in the initial version, we didn’t sufficiently argue why the gauge equivariant convolution is also equivariant to global transformations. Hence, we added a new appendix to the revised paper which argues why the network is based on the intrinsic structure and thus insensitive to the global translation and rotations of the vertex coordinates. Furthermore, we added in the revised version a novel proof that whenever the mesh has an orientation-preserving isometry, the network is equivariant to the signal being moved around by such transformations. As an implication of this, when our method is applied to a grid graph, the network has the same equivariance properties of a group equivariant CNN.
> * The reviewer asks a great question how equivariance affects the universality of the network. We do not have a universality proof for our network, but the literature on equivariant networks on sets (Maron et al. 2019 “Invariant and Equivariant Graph Networks”) and point-clouds (Dym & Maron 2020, “On the Universality of Rotation Equivariant Point Cloud Networks”) suggest that such networks are universal when the intermediate activations contain high-order representations, which is possible in our method. Proving universality for our method in an interesting direction for future work.
> * “K_neigh dependent on r”. Please see our response to all reviewers jointly.
> * “How efficient is the non-linearity?” Thank you for this comment. We have added a discussion about computational complexity in the relevant section in the revised version.
> * “In the MNIST experiment, pooling is applied (appendix D.1). How does this interfere with the equivariance property?” Pooling is done by strides. This respects gauge equivariance, as well as equivariance to rotations on the flat mesh, which has a rotational isometry. However, as always with CNNs with stride, equivariance to single pixel translation is lost.
> * Thank you for spotting the typo.

---

> ### Author Response · Authors · 2020-11-25
> **Added equivariance testing and analsysis of the computational cost of non-linearity**
>
> We have uploaded a new revision that implements two of your great suggestions for new experiments. In appendix F.2, we empirically analyse the equivariance properties of the model with respect to three kinds of transformations. In appendix F.1, we empirically evaluate the cost of the DFT in the non-linearity.

---

### Official Review · AnonReviewer2 · 2020-11-03
**Good idea presented in an unnecessarily complex fashion.**

**Rating:** 7
**Confidence:** 3

**Review:**

This paper presents Gauge Equivariant Mesh CNNs. The method is motivated by the fact that graph convolutions can be modified for meshes to take into account the angular arrangement of local neighborhoods. The result is a Mesh-CNN that is equivalent to GCNs with anisotropic gauge equivariant kernels.

STRENGTHS

- The problem tackled here is very important and well motivated. The authors identify the issues related to existing networks and devise a sensible approach.
- The approach is detailed and carefully patches the problems in mesh convolutions.
- Introducing non-linearity to such networks is far from being trivial. I thank the authors for conveying an analysis on this front.

WEAKNESSES

- I believe that this paper (as well as many other prior works) are missing an important connection to the rich literature of 3D vision. From Ajmal Mian's work to Spin images, from SHOT to the recent deeply learned PPF-FoldNet (and point pair features thereon), there are just too many works that compute local reference frames to canonically orient the local geometry. Many of these works compute some form of a point cloud normal (thus a tangent plane) and choose a direction orthogonal to it (hence fix a gauge). Traditionally, this is known as 'local reference frame' and have set an important milestone in 3D descriptors, learned or not. Simply changing the terminology and rephrasing the established results (e.g. connection of local frames via a rotation) through the lense of Riemannian geometry should not distinguish this paper. I like this principled explanation (only a personal preference) but the ties to all those works should be made concrete. For example, recently, Zhao et al. have used local reference frames on meshes and point clouds to design equivariant point cloud networks:
* Zhao, Y., Birdal, T., Lenssen, J. E., Menegatti, E., Guibas, L., & Tombari, F. (2020). Quaternion Equivariant Capsule Networks for 3D Point Clouds. European Conference on Computer Vision (ECCV)

- It would then be nice to mention clearly that one would like to omit certain local reference frame choices and directly convolve as there is an ambiguity in the tangent plane.

- There are now many works which can exploit Riemannian geometry to craft convolution operators. Technically, there are subtle differences which makes the contributions of the paper less clearer. I would strongly suggest that the paper is revised such that the contributions are stated very clearly. Otherwise, it is hard to figure out which parts already existed and which are novel.

- I consider Sec. 6.1 to be a synthetic dataset because changing the underlying lattice for MNIST does not have practical use. And the results presented in Tab. 2 have some interesting accuracies such as 1.40. Why is that so low? Can we just fix this by picking a naive baseline: Choose an LRF (see above, use SHOT's frame for instance), canonicalize the points locally and apply convolution. How would this simple approach perform? I think some similar idea is already mentioned in:
* Yang, Z., Litany, O., Birdal, T., Sridhar, S., & Guibas, L. (2020). Continuous Geodesic Convolutions for Learning on 3D Shapes. arXiv preprint arXiv:2002.02506.

- Would be possible to briefly summarize Mesh-CNN? This paper gives an improvement over that so it could be beneficial to see (even in a supplementary) how it is improved. Or is Mesh-CNN is used just to refer to the category of works?

- There is a form of 'discretization' of the parallel transport going on. Can't we compute the vector heat to do a better and faster discretization:
* Sharp, Nicholas, Yousuf Soliman, and Keenan Crane. "The vector heat method." ACM Transactions on Graphics (TOG) 38.3 (2019): 1-19.
In general it would be great to have an understanding of the proposed transport operator.

- Some more focused explanation of the representation theory could be great.

My recommendation is positive because the community needs principled ways of convolution on non-Euclidean domains, and this paper seems to make an incremental contribution towards that direction. However, lack of thorough evaluations, the missing links to the literature and the rather inaccessible presentation of the material should definitely be improved.

---

> ### Author Response · Authors · 2020-11-20
> **Thank you for the review and helpful suggestions**
>
> We thank the reviewer for their thorough review. We’d like to respond to the following points:
> * We fully agree with the reviewer that the related work lacked mention of (equivariant) point-cloud methods, such as Zhao (2020) and PPF-FoldNet. The main difference between our work and these two papers (or for example Tensor Field Networks) is that our method is an intrinsic 2D method, meaning it is dependent only on the intrinsic curvature of the 2D manifold, not on the embedding of the manifold in R^3. Hence, the kernel is in general dependent on only the radius and angle of the neighbours in the tangent plane (we do not use the radius in our implementation). These point-cloud methods on the other hand, use 3D features (or 4D for PPF-FoldNet) and kernels defined in this higher-dimensional space. To give an example of where the difference becomes obvious, consider a plane and a cylinder. Locally, both have the same intrinsic 2D geometry, so our convolution behaves similarly on both. The extrinsic embedding of the manifold of these differs, however, so point-cloud methods treat these differently. Which paradigm is more desirable, is dependent on the application. We have added a paragraph regarding the relation of our work to these methods in the related work of the revised version. Furthermore, we have added a new appendix “Equivariance” that includes a discussion about the fact that our method only depends on intrinsic properties of the manifold.
> * “Would be possible to briefly summarize Mesh-CNN?” In line with the above, we consider a Mesh-CNN a method that intrinsically operates on the discretised 2D manifold.
> * The reviewer asks what would happen when one chooses a local reference frame using a heuristic, e.g. using a SHOT frame, and then just an unconstrained (non-gauge equivariant) convolution kernel. Such an approach has two disadvantages. (1) On some smooth manifolds it is impossible to canonically continuously define a frame everywhere. For example, on the sphere this is impossible by the hairy ball theorem, or on a plane it is impossible to canonically disambiguate the X axis from the Y axis. When discretizing such a manifold to a mesh, the heuristically chosen frame, and thus the convolution output, becomes arbitrarily dependent on the discretisation, which seems undesirable. (2) Equivariant methods are known to be more data efficient. This may also apply in our case, as it corresponds to a virtual data augmentation in which in all layers, the convolution input is rotated.
> * “There is a form of 'discretization' of the parallel transport going on. Can't we compute the vector heat to do a better and faster discretization”. Please see our joint response to all reviewers.
> * “Some more focused explanation of the representation theory could be great.” We agree that the representation theory can be hard to grasp from this paper alone and have attempted to improve the explanation of representation theory in section 3. However, a paper can not substitute a proper textbook, to which we added a reference in the paper.
> * “I consider Sec. 6.1 to be a synthetic dataset because changing the underlying lattice for MNIST does not have practical use.” We agree that this experiment is not practically relevant. The reason we include it, is because it demonstrates an important property of the method: it is invariant to isometric transformations (as discussed in the new appendix “Equivariance”), but not invariant to transformations that stretch the surface.

---

> ### Author Response · Authors · 2020-11-25
> **Implemented proposed baseline**
>
> We have uploaded a new revision that includes the baseline you propose: a non-equivariant model with SHOT-based local reference frames. Thank you for this suggestion!

---

### Official Review · AnonReviewer5 · 2020-11-10
**Authors address a crucial problem with graph convolutions on meshes: the isotropy of the kernel that clearly limits the deformation of the mesh. If anisotropic kernels are  introduced than convolutions have to be equivariant with respect to the gauge and this is what this paper achieves with parallel transport.**

**Rating:** 9
**Confidence:** 4

**Review:**

Graph convolution has been defined to be permutation equivariant to the neighborhood vertices. If one were to define an anisotropic kernel then a reference edge would have to be defined corresponding to edge angle 0. Figure 1 is very illuminating.

Equivariance with respect to this reference can be achieved only with a special mechanism. The authors here proposed message passing via edge transporters.
The crux of the approach is in equation (2) and in particular in $\rho(g_{p \rightarrow q} \in [0,2\pi)$. What happens is that the feature vectors of the adjacent vertices have to be transported to the center node, namely a transformation from the local coordinates in p to the local coordinates in q correcting, thus, the underlying gauge difference. This step includes also an alignment of the tangent planes. I strongly believe that appendix A belongs to the main paper.

The second crucial point is that equivariance imposes a linear constraint on the kernels $K_{Self}$ and $K_{Neigh}$. This allows a kernel to be written as a linear combination of 20 kernels for $K_{Neigh}$ and $K_{Self}$ resulting in 24 only unknowns for one layer.

The related work section is comprehensive.

The experiment on shape correspondence achieves performance comparable to the state of the art spiralNet++. The paper would benefit from other experiments on manifold like performing mesh convolutions for human or object reconstruction from images.
Faust is pretty standard in the GML community but it is an easy task in terms of feature learning.

The paper contribution is elegant and significant: Gauge equivariance  is a necessity if you want an anisotropic diffusion.

The paper is unreadable without the appendix and somehow it would be better to make it self-contained and move the experiments in the appendix.

---

> ### Author Response · Authors · 2020-11-20
> **Thank you for the review and helpful suggestions**
>
> We thank the reviewer for their kind words and constructive suggestions. We agree that the geometric computation is an important part of the method and thus include a short version of appendix A in the main paper, in order to make it more self-contained.

---

### Author Response · Authors · 2020-11-20
**Joint response to all reviewers**

We thank all the reviewers for their thorough reviews and helpful suggestions. First, we’d like to respond to some common points:
* Reviewer 2 asks about computing the geodesic quantities using the Vector Heat Method (VHM) and reviewer 4 asks about how our geometric pre-computation pipeline differs from that of prior methods. Both reviewers here refer to an important design choice we made. What we do is to restrict ourselves to message passing over geodesics that span one edge. For these geodesics, we use exact parallel transport and logarithmic maps, which can be computed locally using simple linear algebra, as shown in appendix A. We omit messages that pass along geodesics over faces or multiple edges, for which a method like the VHM can compute parallel transport and logarithmic maps. This decision was made for three reasons: (1) by passing messages only over edges and not over longer geodesics, we try to imitate the highly successful 3x3 kernels used in planar images, (2) the implementation simplifies by having to solve only local linear computations at each vertex and not having to solve global problems, which is required for VHM or other forms of finding geodesics, (3) using only edges, each point has fewer neighbours, which dramatically speeds up the convolution.
* Reviewers 1 and 4 ask why our kernel is not dependent on the radial distance to the neighbour. It is totally possible to do so and our analysis of the kernel constraint directly translates to this case. For such a kernel, one has K(r, theta)=F(r)K_neigh(theta), where the first factor is unconstrained and the second factor is the same as in our discussion. We added a footnote on this. We experimented with this, and found it advantageous to use kernels that do not depend on the radius. We suspect this is the case because we only pass messages over direct edges, so that for each point p, and each angle theta, there is at most one neighbour at on radius r.

Incorporating all comments, we have uploaded a revised version of the paper with the following changes:
* We have included a summary of Appendix A in the main paper, in order to make the main paper more self-contained, as suggested by reviewer 5.
* We have added a reference to point-cloud methods in the related work in response to the comment by reviewer 2.
* We have attempted to clarify the discussion of representation theory in section 3, as suggested by reviewer 2.
* We have added a new appendix discussing the equivariance properties of the model, as suggested by reviewer 1. Furthermore, we added a new proof that the model is equivariant to isometries of the mesh.
* We comment on the computational complexity of the regular non-linearity, as suggested by reviewer 1.
* We comment on the adding the radius as a parameter to the kernel in a footnote, as suggested by reviewers 1 and 4.

We are working on a final revision that will include new empirical results, as requested.

---

> ### Author Response · Authors · 2020-11-25
> **New revision with equivariance testing, non-equivariant baseline and analysis of computational cost**
>
> We have uploaded a new revision with the following additions:
> * As requested by Reviewers 1 and 4, we have added an extensive appendix F.2, which shows empirically the equivariance properties of the model and the non-linearity.
> * As requested by Reviewer 2, we have added a non-gauge-equivariant baseline, which used local reference frames based on SHOT, to the FAUST experiment.
> * As requested by Reviewer 1, we have added appendix F.1, which shows the run-time cost of the model and the RegularNonLinearity with varying number of samples.

---

### Decision · Program_Chairs · 2021-01-07
**Final Decision**

**Decision:**

Accept (Spotlight)

**Comment:**


This paper addresses a crucial problem with graph convolutions on meshes.
The authors identify the issues related to existing networks and devise a sensible approach.
The work presents a novel message passing GNN operator for meshes that is equivariant under gauge transformations.
The reviewers unanimously agree on the both the importance of the problem and the impact the proposed work could have.

Suggestions for next version:
- The paper is unreadable without the appendix and somehow it would be better to make it self-contained
- Additional references as suggested in the reviews.
- Expanded experiments as suggested by R4, will also improve reader's confidence in the method.

I would recommend acceptance. I would request the authors to release a sufficiently documented and easy to use implementation. This not only allows readers to build on this work but also increase the overall impact of this method.